# Finding Flatter Minima with SGD

**Stanisław Jastrzębski**[1,2,*], **Zachary Kenton**[2,*], **Devansh Arpit**[2], **Nicolas Ballas**[3], **Asja Fischer**[4], **Yoshua Bengio**[2] & **Amos Storkey**[5]

[1] Jagiellonian University, Cracow, Poland
[2] MILA, Université de Montréal, Canada
[3] Facebook, Montreal, Canada
[4] University of Bonn, Bonn, Germany
[5] The University of Edinburgh, United Kingdom
[*] Equal Contribution

## Abstract

It has been discussed that over-parameterized deep neural networks (DNNs) trained using stochastic gradient descent (SGD) with smaller batch sizes generalize better compared with those trained with larger batch sizes. Additionally, model parameters found by small batch size SGD tend to be in flatter regions. We extend these empirical observations and experimentally show that both large learning rate and small batch size contribute towards SGD finding flatter minima that generalize well. Conversely, we find that small learning rates and large batch sizes lead to sharper minima that correlate with poor generalization in DNNs.

## 1 Introduction

Deep neural networks (DNNs) have demonstrated good generalization ability and achieved state-of-the-art performances in many application domains despite being massively over-parameterized (Zhang et al., 2016). But the reason behind their success remains an open question. It has been argued that flatter minima tend to generalize better than sharper ones (Hochreiter & Schmidhuber, 1997; Shirish Keskar et al., 2016). Shirish Keskar et al. (2016) empirically show that larger batch-size correlates with sharper minima and that networks trained with a large batch size usually have worse generalization performance than those trained with small batch-size. Extending this claim, we show that both large learning rate and small batch size lead to flatter minima and that empirically these flatter minima have better generalization compared with sharper ones. Thus our claim is in support of the findings of Goyal et al. (2017); Hoffer et al. (2017).

## 2 Experiments

We begin by investigating the impact of learning rate $\eta$ and batch-size $S$ on the flatness of regions that SGD ends in (measured by the Hessian of the DNN loss with respect to its parameters). For this experiment we use a 4-layer batch-normalized rectified linear unit (ReLU) multi-layer perceptron (MLP) and train it on the Fashion-MNIST dataset (Xiao et al., 2017). We specifically study how the ratio of learning rate to batch size ($\frac{\eta}{S}$) impacts the flatness of region that SGD finds and record the validation accuracy of the corresponding model. To measure the flatness at the end of training, we compute both the spectral norm (largest eigenvalue denoted by $\max_j \lambda_j$, where $\lambda_j$ are the eigenvalues of the Hessian) and the Frobenius norm (denoted by $\|H\|_F$) of the Hessian $H$. Notice a higher value of the spectral norm and Frobenius norm (at minima) implies a sharper region. We train multiple separate models with a combination of learning rate $\eta \in [5e-3, 1e-1]$ and batch size $S \in [25, 1000]$. Each experiment is run for 200 epochs; most models reach approximately $100\%$ accuracy on the training set.

The results are shown in figure 1a and figure 1b. We find that as the ratio $\eta/S$ grows, both the spectral and Frobenius norm of the Hessian at the final model parameter decrease, suggesting that *higher learning rate to batch size ratio pushes the optimization towards flatter regions.*

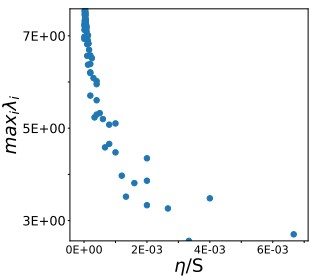
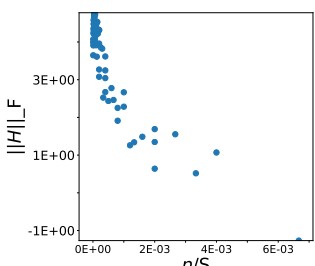
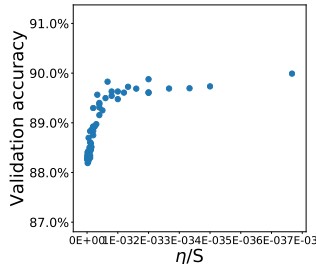

(a) Correlation of $\frac{\eta}{S}$ with largest eigenvalue of the Hessian.

(b) Correlation of $\frac{\eta}{S}$ with logarithm of Frobenius norm of the Hessian.

(c) Correlation of $\frac{\eta}{S}$ with validation accuracy.

Figure 1: Impact of SGD with ratio of learning rate $\eta$ and batch-size $S$ on flatness of final minima and validation accuracy for a 4 layer ReLU MLP architecture on FashionMNIST dataset.

Figure 1c shows the results from exploring the impact of $\frac{\eta}{S}$ on the final validation performance from the above experiments. It suggests that *better generalization correlates with higher learning rate to batch size ratio.*

Thus together, figures 1b and 1c show that a large ratio of learning rate and batch size guides SGD towards flatter regions and parameters in such regions roughly correlate with better generalization.

Now we qualitatively compare the flatness of minimum found by SGD using different ratios of learning rate to batch size. To investigate this behavior of SGD, we train three Resnet-56 models He et al. (2016) on CIFAR-10 using SGD with different values of $\frac{\eta}{S}$. Our baseline model uses $\frac{\eta=0.1}{S=128}$. In comparison, we investigate a large batch model with $\frac{\eta=0.1}{S=1024}$ and a small learning rate model with $\frac{\eta=0.01}{S=128}$, which have approximately the same $\frac{\eta}{S}$ ratio. We follow (Goodfellow et al., 2014) by investigating the loss on the line interpolating between the parameters of two models. More specifically, let $\vec{\theta}_1$ and $\vec{\theta}_2$ be the final parameters found by SGD using different $\frac{\eta}{S}$, we report the loss values $L((1-\alpha)\vec{\theta}_1 + \alpha\vec{\theta}_2)$ for $\alpha \in [-1, 2]$. *Results indicate that models with larger batch size (Fig. 2(a)-right minimum) or lower learning rate (Fig. 2(b)-right minimum) end up in a sharper minimum relative to the baseline model (Fig. 2(a,b)-left minimum).*

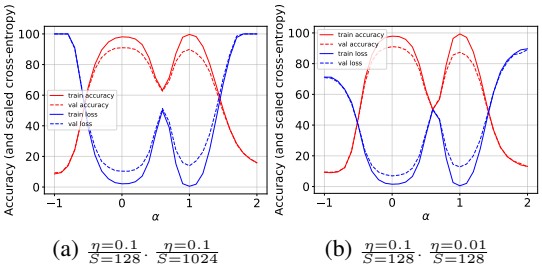

(a) $\frac{\eta=0.1}{S=128}$. $\frac{\eta=0.1}{S=1024}$

(b) $\frac{\eta=0.1}{S=128}$. $\frac{\eta=0.01}{S=128}$

Figure 2: Interpolation results for Resnet-56 networks trained with different ratio of learning rate to batch-size $\frac{\eta}{S}$. The symbol $\alpha$ (x-axis) corresponds to the interpolation coefficient. Higher $\frac{\eta}{S}$ ratio leads to wider regions. Subcaptions give $\eta, S$ at $\alpha = 0, 1$.

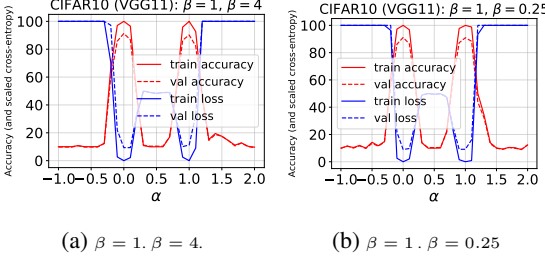

(a) $\beta = 1$. $\beta = 4$.

(b) $\beta = 1$. $\beta = 0.25$

Figure 3: Interpolation results for VGG-11 networks with the same learning rate to batch-size ratio: $\frac{\eta=0.1\times\beta}{S=50\times\beta}$, but different $\beta$. Identical noise levels are qualitatively similar. Subcaptions give $\beta$ at $\alpha = 0, 1$.

Next we train VGG-11 models (Simonyan & Zisserman, 2014) on CIFAR-10, such that all the models are trained with the same ratio $\frac{\eta}{S}$ of learning rate to batch size but with different individual values of $\eta$ and $S$. Specifically, we use $\frac{\eta=0.1\times\beta}{S=50\times\beta}$, where we set $\beta = 0.25, 1, 4$. We then interpolate between the model parameters found when training with $\beta = 1$ and $\beta = 4$ (Fig. 3-left), and $\beta = 1$ and $\beta = 0.25$ (Fig. 3-right). The interpolation results indicate that all the

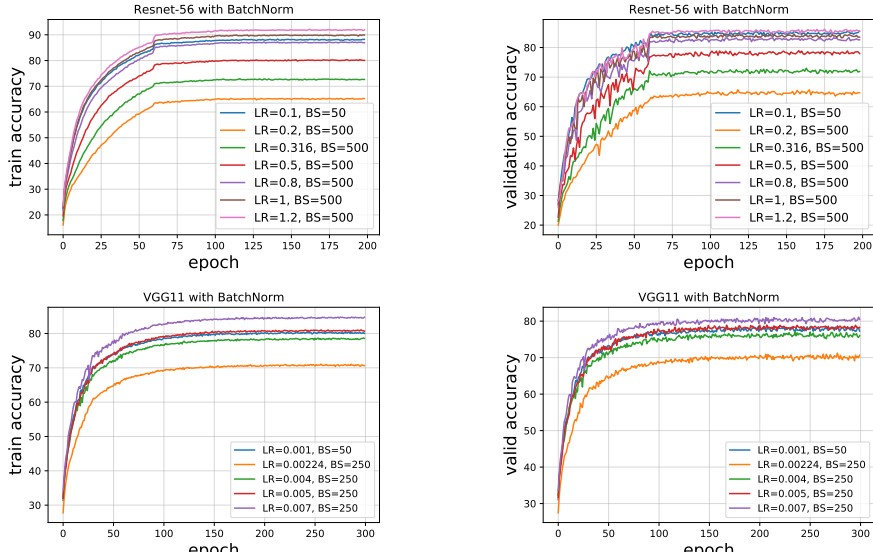

Figure 4: ResNet (top) and VGG11 (bottom) on CIFAR10. Rescaling the learning rate to reproduce a similar learning curve when going from a small batch-size (blue) to a large one. In both experiments rescaling learning rate by same amount as batch size gives closer match than rescaling by the square root of batch size.

minima have similar width, suggesting that *for the same ratio of learning rate to batch size, SGD ends up in regions of similar flatness.*

Note that these experiments were repeated several times using both architectures with different random initializations and qualitatively similar plots were achieved.

In all our above analysis, we use arbitrarily learning rate and batch size in a linear ratio while making our claim that the flatness of minima (and generalization) correlates with this ratio. Now we try to further investigate if this ratio controls the dynamics of learning in SGD. To do so, we train a baseline Resnet-56 model[1] with learning rate $0.1$ and batch size $50$. Then we experiment with different models trained with a fixed batch size of $500$ (10 times larger), but with learning rates in $\{0.2, 0.316, 0.5, 1, 1.2\}$ (notice $0.316$ is $\sqrt{(10)}$). The results are shown in figure 4 (top). The goal of this experiment is to investigate which learning rate best matches the baseline training dynamics when the batch size is multiplied by a certain factor. We perform a similar experiment with the VGG-11 architecture[2] shown in figure 4 (bottom). *We find that the learning dynamics of models trained with similar ratios of learning rate to batch size are roughly quite similar both in terms of training and validation accuracy.* But we do not expect this result to hold when multiplying learning rate with a very large factor because it may cause divergence.

## 3 CONCLUSION

We empirically find that using large learning rate and/or small batch size steers SGD towards flatter minima. We verify this flatness both quantitatively (using Hessian) and qualitatively (by interpolating the DNN loss between minima achieved by different learning rates and batch sizes). Additionally we also find that such flatter regions correlate with better generalization performance. Finally, we show that models trained with similar ratio of learning rate to batch size both end up in minima of qualitatively similar flatness, and have similar learning dynamics.

---

[1] $\eta$ drops by a factor of 10 on epochs 60, 100, 140, 180.

[2] $\eta$ drops by a factor of 2 every 25 epochs.

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
