# OpenReview forum: "Finding Flatter Minima with SGD"
_ICLR.cc/2018/Workshop — Accept_

### Official Review · AnonReviewer5 · 2018-02-23
**A poster I would be happy to visit at ICLR**

**Rating:** 8
**Confidence:** 4

**Review:**

The whole "flat minima" topic in deep learning is being actively debated, without much hard data. I appreciate the authors doing these experiments and sharing them with the community. I would be happy to come and visit a poster about this experiment, and think it would add to ICLR.

One question/comment on the content: you show a plot of the largest eigenvalue of the Hessian. But I can manipulate it by multiplying the cost function by 100 (and correspondingly decreasing the step size by 100 to maintain stability). Wouldn't it be more interesting to plot the condition number of the Hessian (i.e., the ratio of the largest to smallest eigenvalue?)

---

### Official Review · AnonReviewer4 · 2018-03-04
**Interesting study on the joint effect of learning rate and batch size.**

**Rating:** 6
**Confidence:** 4

**Review:**

This paper investigates the joint impact of learning and batch size on the optimization of feed-forward neural networks. Previous related work studied the effect of batch size alone, without considering the learning rate in such detail. The paper is clearly written, with interesting findings such as "better generalization (and flatter region of convergence) correlates with higher learning rate to batch size ratio", and that "similar ratio yield similar learning dynamics". Below are some suggestions that I think would make these claims more convincing. For future work, it would be interesting to see if these insights generalize to other model architectures (e.g. recurrent networks, attention models).

Some notes for the authors:
1. The second claim (Figures 2 and 3) would be more convincing if the architectures remain unchanged. The authors do mention that both architectures yielded similar results, but the presentation of two architectures as opposed to a single one for this specific study is distracting.
2. There are some minor formatting issues. Figures 2 and 3 are too small and not legible at normal resolutions. The number sqrt(10) is not formatted correctly (see https://tex.stackexchange.com/questions/167892/square-root-radical-sign).
3. For the last claim (on learning dynamics), it would be more convincing had the authors shown similar results for multiple baselines as opposed to a single one (learning rate=0.1/batch size=50).

---

### Official Review · AnonReviewer3 · 2018-03-11
**Some insights from experimental results, however, there is limited novalty**

**Rating:** 5
**Confidence:** 4

**Review:**

This paper empirical studies the relationship among the mini-batch size, learning rate, the sharpness of the local minima and the final performance. In general, it shows that large learning rate and small batch size would usually lead to better generalization performance. The experiment is insightful and expected. There is not much theoretical analysis about any of these observations. Additionally, most of these relationships are known from previous work.

Detailed comments:

(1) The author should mention some highly related work that studies similar topic, such as
    [1]Coupling Adaptive Batch Sizes with Learning Rates.Lukas Balles, Javier Romero, Philipp Hennig
    [2] A Bayesian Perspective on Generalization and Stochastic Gradient Descent, Samuel L. Smith, Quoc V. Le

(2)  Why does the work use spectral norm and Frobenius norm of Hessian to measure the flatness? Why not use the conditioned number (the ratio between largest eigenvalue and smallest once) of the Hessian would be a better metric?

(3) In Figure 4, the gap between training accuracies for various setting is very large. For example, "LR = 0.2, BS = 500" leads to about 65% accuracy. "LR=0.5, BS=500" causes about 80% accuracy. I wonder whether such big differences can be expected in general.

---

### Decision · Program_Chairs · 2018-03-20
**ICLR 2018 Workshop Acceptance Decision**

**Decision:**

Accept

**Comment:**

Congratulations, your paper was accepted to the ICLR workshop.